# Characterizing Growing Season Length of Subtropical Coniferous Forests with a Phenological Model

Yuan Gong [1,2], Christina L. Staudhammer [2,*], Susanne Wiesner [2,3], Gregory Starr [2] and Yinlong Zhang [1,*]

1    Co-Innovation Center for Sustainable Forestry in Southern China, College of Biology and the Environment, Nanjing Forestry University, Nanjing 210037, China; ygong10@ua.edu
2    Department of Biological Sciences, The University of Alabama, Tuscaloosa, AL 35487, USA; susanne.wiesner@usda.gov (S.W.); gstarr@ua.edu (G.S.)
3    Department of Biological Systems Engineering, University of Wisconsin, Madison, WI 53706, USA
\*    Correspondence: cstaudhammer@ua.edu (C.L.S.); zylnjfu@gmail.com (Y.Z.)

**Abstract:** Understanding plant phenological change is of great concern in the context of global climate change. Phenological models can aid in understanding and predicting growing season changes and can be parameterized with gross primary production (GPP) estimated using the eddy covariance (EC) technique. This study used nine years of EC-derived GPP data from three mature subtropical longleaf pine forests in the southeastern United States with differing soil water holding capacity in combination with site-specific micrometeorological data to parameterize a photosynthesis-based phenological model. We evaluated how weather conditions and prescribed fire led to variation in the ecosystem phenological processes. The results suggest that soil water availability had an effect on phenology, and greater soil water availability was associated with a longer growing season (LOS). We also observed that prescribed fire, a common forest management activity in the region, had a limited impact on phenological processes. Dormant season fire had no significant effect on phenological processes by site, but we observed differences in the start of the growing season (SOS) between fire and non-fire years. Fire delayed SOS by 10 d ± 5 d (SE), and this effect was greater with higher soil water availability, extending SOS by 18 d on average. Fire was also associated with increased sensitivity of spring phenology to radiation and air temperature. We found that interannual climate change and periodic weather anomalies (flood, short-term drought, and long-term drought), controlled annual ecosystem phenological processes more than prescribed fire. When water availability increased following short-term summer drought, the growing season was extended. With future climate change, subtropical areas of the Southeastern US are expected to experience more frequent short-term droughts, which could shorten the region's growing season and lead to a reduction in the longleaf pine ecosystem's carbon sequestration capacity.

**Keywords:** ecosystem physiology; eddy covariance; gross primary production; longleaf pine; phenology; prescribed fire

## 1. Introduction

Forested ecosystems play an important role in the global carbon cycle [1,2], and are among the largest terrestrial carbon sinks [3,4]. It has been hypothesized that forests may have the ability to increase their carbon sequestration capacity as a means to mitigate rising atmospheric $CO_2$ concentrations [5,6]. This increase will likely be through phenological changes, as future model projections indicate that by the end of this century, the growing season length will increase by 29–43 days [7]. Moreover, modifications in forest phenology could have the potential to influence regional-scale weather patterns as well as global climate, through changes in seasonal patterns of short- and long-wave radiation absorption, water vapor and carbon dioxide fluxes, and biogenic volatile organic compounds [8]. Phenological processes of forests vary with climate, geography, weather events and human activities [9–11], yet terrestrial biosphere models do not adequately represent this variability

in phenology [8]. In addition, there is a great deal of uncertainty in how global change will alter the timing of phenological events in many ecosystems and thus, increased uncertainty in how forest carbon cycles will be altered [8,12]. Therefore, knowledge regarding phenological characteristics of different forests, specifically the length of the growing season, can contribute to a better understanding of the terrestrial carbon cycle with global change [13].

The scientific community has used two primary methods for studying phenology. Land surface phenology (LSP) includes near-surface and high-altitude methodologies [13], and ground observed phenology (GP) [14,15]. Early research using LSP methods relied on optical satellite data (remote sensing, RS) such as MODIS and SPOT vegetation indices (leaf area index, enhanced vegetation index, etc.) [16,17]. Observations taken at 3 or 8 d intervals are used to determine patterns of plant growth through statistical or physiological models and phenological time nodes (e.g., carbon uptake period, growing season, etc.) are extracted [18–20]. However, RS observations are limited by cloud cover, rain, or fog [21], and it is difficult to obtain adequate plant phenology data at different ecological (species, populations, communities, ecosystems, etc.) or temporal scales, due to limits in resolution [22,23]. GP methods provide the basis for plant phenology studies at community- and ecosystem-scales, which usually utilize continuous images taken by phenology cameras [14,15]. However, the area captured in these images is rather small and limits our ability to scale to larger areas and/or capture heterogeneity across the landscape.

Phenological models provide a tool to enhance our understanding of changes in phenology. A number of parametric phenological models are currently available, including the seven-parameter double logistic function developed by Gonsamo et al. [19] and four-parameter piecewise logistic functions developed by Zhang et al. [24]. These two models are widely used to fit RS and ground observed data and extract key phenology nodes associated with growing season length [19,20]. These nodes include the start of growing season (SOS), end of growing season (EOS) and length of growing season (LOS = number of days between SOS and EOS; [19,20]), which are estimated via threshold or third derivative methods [19,20,25]. A nine-parameter phenology model developed by Gu et al. [26] also uses a logistic function and has been found to have higher accuracy in predicting plant growing season in warmer regions and urban forests [26–30]. However, relatively fewer studies have parameterized this model to test hypotheses about plant phenological processes [29].

While these logistic models are usually parameterized with ground-observed ecosystem productivity, camera-based indices, and RS-predicted vegetation index at different ecological scales [8,31], the growing availability of eddy covariance (EC) data, which captures direct measurements of $CO_2$ fluxes from whole ecosystems, has led to their widespread use in parameterizing phenological models [20]. Estimates of gross primary production (GPP) are then fit to these logistic functions to obtain dynamic representations of important phenology nodes, such as the start, end, and overall length of the growing season [18,26]. This improved understanding of the growing season of forests using GPP can aid the scientific community's understanding of the dynamic characteristics of LSP at different scales, which may improve our estimates of regional carbon budgets with climate change [20]. However, past research linking phenological processes in forests using EC technology has been mainly concentrated in temperate regions, with a focus on the effects of geographic and environmental controls, such as latitude/longitude, air/soil temperature, precipitation, and radiation [32]. While subtropical regions also have extensive forested areas, EC-based phenological studies in these regions have received less attention [28,33].

In addition, relatively few studies have examined the complex interactions between weather and prescribed fire, a common forest management activity in many regions, both of which may control forest phenology and ultimately their system's carbon dynamics [34]. Prescribed fire is used extensively in the southeastern US to maintain native biodiversity and reduce the risk of wildfire; however, prescribed fire frequency and intensity may change with global climate change due to the forecasted increased drought conditions and

lengthened growing season [35]. In light of the potential for phenology to be impacted by climate change-fire interactions, there is a need for a better understanding of the mechanisms driving changes in phenological processes in subtropical forests.

We used long-term, EC-measured $CO_2$ flux and micro-meteorological data in three subtropical coniferous forests [36], to study phenological characteristics and responses to prescribed fire, weather events, and variation in air temperature using a photosynthesis-based phenology model. This study addressed the following hypotheses: (1) Soil water availability will significantly affect ecosystem-scale phenological processes, i.e., higher soil water availability will be associated with a longer growing season. (2) Phenological nodes describing the growing season length (SOS, EOS, and LOS, etc.) for the sites will be affected by climate (e.g., air temperature, precipitation, and radiation) and prescribed fire, i.e., increased air temperature will prolong LOS (early SOS and late EOS), while drought will lead to a shortening of the growing season. (3) Prescribed fire during the dormant season will delay the SOS, and this will be compounded if water availability is limited during the recovery period. By testing these hypotheses, we will increase our understanding of subtropical forest phenological processes and their response to climate change and weather anomalies, as well as provide a basis for carbon budget planning and mitigation of global change.

## 2. Materials and Methods

### 2.1. Study Area

The study was conducted at the Jones Center at Ichauway (JCI) in southwestern Georgia, USA, using data from three sites collected during 2009–2017. The 11,000-hectare JCI property (31.22° N, 84.47° W; Figure 1) has a relatively flat terrain with long-term average annual precipitation of 1310 mm; air temperature ranges from −3 °C to 23 °C in winter, and 16 °C to 31 °C in summer [37]. The regional climate is classified as subtropical. All three sites are maintained by frequent low intensity fire on a two-year return interval (odd-numbered years; Table 1) [36]. Due to the differences in soil water holding capacity and land use legacy, the sites have different understory vegetation composition (Table 1) [38]. The mesic site lies on somewhat poorly drained sandy loam over sandy clay loam and clay textured soils [39]. Soils at the intermediate site have a depth to the argillic horizon of approximately 165 cm and are well-drained [39]. The xeric site lies on well-drained deep sandy soils with no argillic horizon [39]. All sites are situated within 10 km of each other (Figure 1), with heights above sea level of 65 m for the mesic site, 55 m for the intermediate site and 60 m for the xeric site. Inside the flux footprint area of the three EC flux towers, long-leaf pine trees (*Pinus palustris* Mill.) are the dominant woody plants, averaging 100 years of age, and understory vegetation includes shrubs such as *Diospyros virginiana* L and *Aristida stricta* Michx [40].

**Table 1.** Description of the three eddy covariance (EC) flux sites at the Jones Center at Ichauway (JCI), Newton, GA, USA.

|  | Mesic | Intermediate | Xeric |
|---|---|---|---|
| DBH (cm) | 25.9 | 42.5 | 22.5 |
| Water holding capacity | 40.3 | 27.6 | 18.5 |
| LAI ($m^2/m^2$) | 2.34 | - | 1.87 |
| NDVI | 0.7 | 0.7 | 0.65 |
| EVI | 0.37 | 0.35 | 0.34 |

DBH: diameter at breast height, LAI: leaf area index (Source: MODIS-MOD15AH2 vegetation product; https://ladsweb.modaps.eosdis.nasa.gov/), NDVI: normalized difference vegetation index, EVI: enhanced vegetation index (Source: [40]). Water holding capacity: cm water per cm soil, measured in upper 3 m of soil.

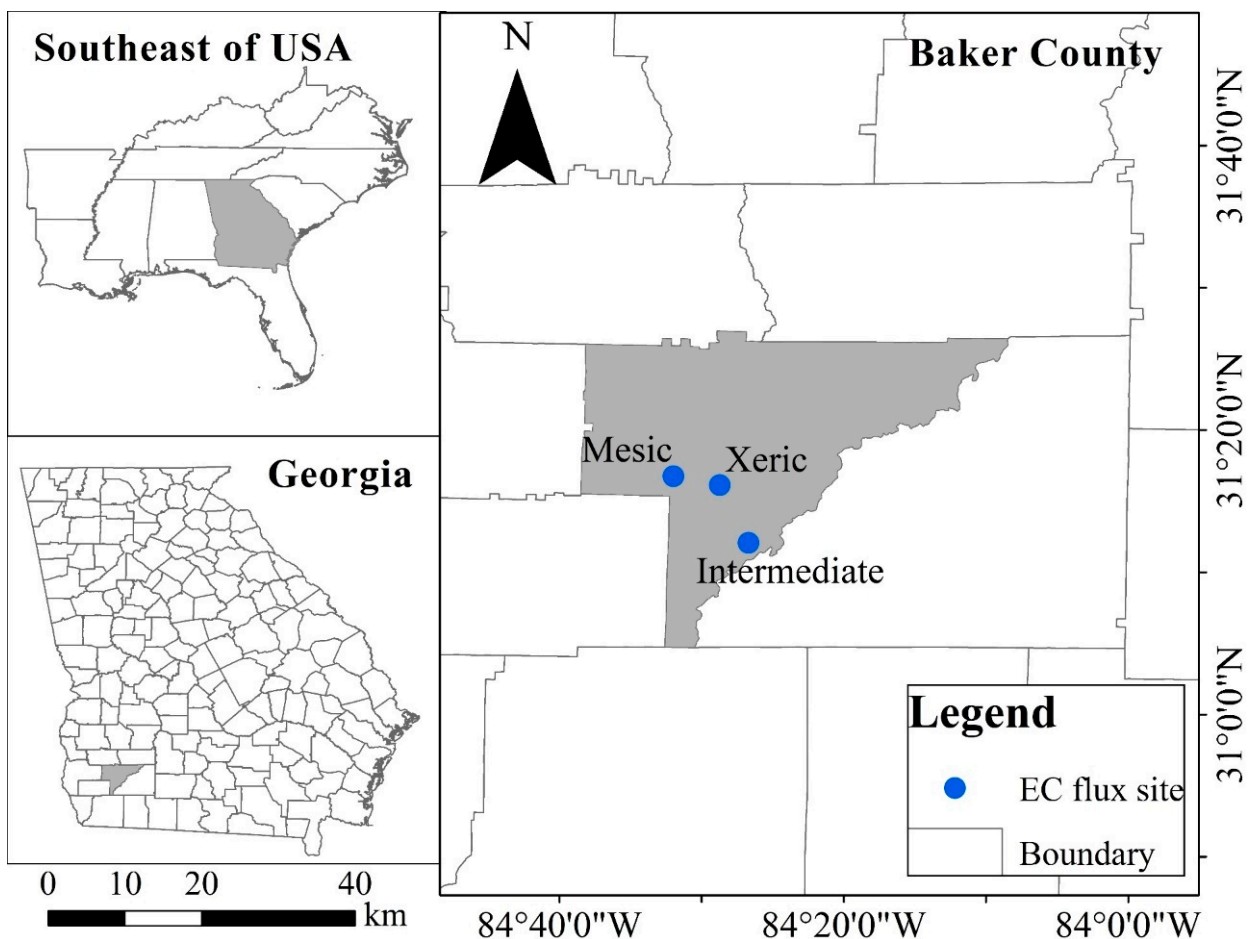

**Figure 1.** Geographical location map of three subtropical coniferous forest EC sites inside the JCI. Administrative boundary data downloaded from: USGS National Boundary Dataset (NBD, [41]), the geographic coordinate system of these map is GCS_WGS_1984 and visualized using ArcGIS version 10.2 (Environmental Systems Research Institute, Redlands, CA, USA).

### 2.2. Net Ecosystem CO₂ Exchange Using Eddy Covariance

Net ecosystem $CO_2$ exchange (NEE; $\mu$mol $CO_2$ m$^{-2}$ s$^{-1}$/g C m$^{-2}$ s$^{-1}$) was measured at all sites with open-path eddy covariance (EC) techniques, measuring at 10 Hz [42] using LI-COR $CO_2$/$H_2O$ infrared gas analyzers (Li-7500, LI-COR, Lincoln, NE, USA) accompanied by three-dimensional ultrasonic anemometers (CSAT-3, Campbell Scientific Instruments, Logan, UT, USA). Each EC measurement system was installed approximately 4 m above mean canopy height [42]. Tower heights were 34.4, 37.5 and 34.9 m, for the mesic, intermediate, and xeric sites, respectively, resulting in a ~600 m upwind fetch of the EC systems at all sites [41].

Micrometeorological data, including photosynthetically active radiation (PAR; LI-190, LI-COR Inc., Lincoln, NE, USA), global radiation (LI-200SZ, LI-COR Inc., Lincoln, NE, USA), incident and outgoing shortwave and longwave radiation to calculate Rn (NR01, Hukseflux Thermal Sensors, Delft, The Netherlands), precipitation (TE525 Tipping Bucket Rain Gauge, Texas Electronics, Dallas, TX, USA), and air temperature ($T_a$) and relative humidity (HMP45C, Campbell Scientific Instruments, Logan, UT, USA) were collected above the canopy and stored on CR-5000 dataloggers (Campbell Scientific Instruments, Logan, UT, USA).

### 2.3. Gross Primary Production Partitioning from NEE

Flux data were processed with the EdiRe software (v.1.4.3.1184) to calculate $CO_2$ flux at 30-min intervals, using a coordinate rotation, frequency response correction, density correc-

tion and spectral attenuation. QA/QC of the $CO_2$ flux data was also maintained by filtering data that did not pass plausibility tests (i.e., NEE < −30 and NEE > 30 µmol $CO_2$ m$^{-2}$ s$^{-1}$), stationarity criteria, and integral turbulent statistics [42].

Missing half-hourly flux data was gap-filled using separate functions for NEE during daytime and nighttime. When photosynthetically active radiation (PAR) was ≥ 10 µmol m$^{-2}$ s$^{-1}$, daytime NEE data were gap-filled using a Michaelis-Menten approach, and when PAR was < 10 µmol m$^{-2}$ s$^{-1}$, nighttime NEE data were gap-filled using a modification of the Lloyd and Taylor approach [43], both on a monthly basis [44]. Where too few observations were available to produce stable and biologically reasonable parameter estimates, annual equations were used to gap-fill daytime and nighttime NEE data by site. Half hourly fluxes of NEE in µmol $CO_2$ m$^{-2}$ s$^{-1}$ were used to calculate gross primary production (GPP) and ecosystem respiration (Re) as follows [42]:

$$Re = GPP + NEE \tag{1}$$

*2.4. Ecosystem-Scale Plant Photosynthetic Phenology Model*

This study used a GPP-based phenology model [26] to analyze the dynamic changes of vegetation phenology in each site from 2009 to 2017, and to discuss the potential impact of disturbance on the ecosystem-scale phenological processes. The GPP-based phenology model was originally developed utilizing interannual daily maximum GPP (µmol $CO_2$ m$^{-2}$ s$^{-1}$) obtained from half-hourly observations to quantify the seasonal change of canopy photosynthetic capacity [26]. Due to the large number of evergreen trees distributed in the flux source area of the sites (Table 1), however, canopy photosynthesis remains active year-round. Even though we have observed significant seasonal variation in GPP derived from 30 min data [44], models of daily maximum GPP did not yield ecologically meaningful estimates of phenology parameters. Therefore, this study applied the GPP-based phenology model using daily cumulative GPP (g C m$^{-2}$ d$^{-1}$).

In the phenological model (Equations (2) and (3)), $A(t)$ represents daily cumulative GPP and $k(t)$ is the GPP growth rate (daily change in GPP), as defined by the $A(t)$ function (g C m$^{-2}$ d$^{-1}$ d$^{-1}$) at day $t$ ($t = 1, \ldots, 365$), estimated for each site by year:

$$A(t) = y_0 + \frac{a_1}{\left[1 + \exp\left(-\frac{t-t_{01}}{b_1}\right)^{c_1}\right]} - \frac{a_2}{\left[1 + \exp\left(-\frac{t-t_{02}}{b_2}\right)^{c_2}\right]} \tag{2}$$

$$k(t) = \frac{dA(t)}{dt} = \frac{a_1 c_1}{b_1} \frac{\exp\left(-\frac{t-t_{01}}{b_1}\right)}{\left[1 + \exp\left(-\frac{t-t_{01}}{b_1}\right)\right]^{1+c_1}} - \frac{a_2 c_2}{b_2} \frac{\exp\left(-\frac{t-t_{02}}{b_2}\right)}{\left[1 + \exp\left(-\frac{t-t_{02}}{b_2}\right)\right]^{1+c_2}} \tag{3}$$

Estimated empirical fitting parameters of $A(t)$ ($a_1$, $a_2$, $b_1$, $b_2$, $c_1$, $c_2$, $t_{01}$, $t_{02}$, and $y_0$) were then used in the $k(t)$ function to calculate the GPP growth rate [26] by site and year. Then, these equations were used to define predicted plant green-up and senescence phases during the growing season: The Recovery Line ($A_{RL}$) and Senescence Line ($A_{SL}$) were defined as Equations (4) and (5) [26], respectively:

$$A_{RL}(t) = k_{PRR}t + A(t_{PRD}) - k_{PRR}t_{PRD} \tag{4}$$

$$A_{SL}(t) = k_{PSR}t + A(t_{PSD}) - k_{PSR}t_{PSD} \tag{5}$$

where: $k_{PRR}$ is the maximum GPP growth rate, which is the maximum recovery rate of GPP, $t_{PRD}$ is the time at which $k_{PRR}$ (day) occurs, $k_{PSR}$ is the minimum GPP growth rate (the maximum GPP senescence rate), and $t_{PSD}$ is the time at which $k_{PSR}$ (day) occurs, which are calculated as follows [26]:

$$t_{PRD} \approx t_{01} + b_1 \ln(c_1) \tag{6}$$

$$t_{PSD} \approx t_{02} + b_2 \ln(c_2) \tag{7}$$

$$k_{PRR} = k(t_{PRD}) \tag{8}$$

$$k_{PSR} = k(t_{PSD}) \tag{9}$$

From the phenology model, maximum GPP is defined as:

$$A_p = \max\{|A(t), t_{start} < t < t_{end}|\} \tag{10}$$

where: $t_{start}$ = 1, and $t_{end}$ = 365 during non-leap years and $t_{end}$ = 366 during leap years. SOS is then defined as the value of $t$ at which GPP predicted by $A_{RL}(t)$ is zero. The start of peak (SOP) of the growing season is the value of $t$ when GPP predicted by $A_{RL}(t)$ reaches $A_p$. The end of peak (EOP) is defined by the date ($t$) when GPP predicted by $A_{SL}(t)$ reaches $A_p$, and EOS is the value of $t$ at which GPP predicted by $A_{SL}(t)$ is zero. Since the study area is located in a warmer subtropical region, the days between SOS and SOP are defined as spring, days between SOP and EOP are defined as summer, days between EOP and EOS are defined as autumn, and days between EOS and the following year's SOS are defined as winter (Figure 2) [26]. The growing season was defined as those days between SOS and EOS, the number of days between SOP and EOP was defined as the length of peak (LOP) [19], and the date of occurrence of $A_p$ was defined as the day of peak (DOP). Computation and visualization of the phenology model was conducted in MATLAB R2014b (MathWorks Inc., Natick, MA, USA).

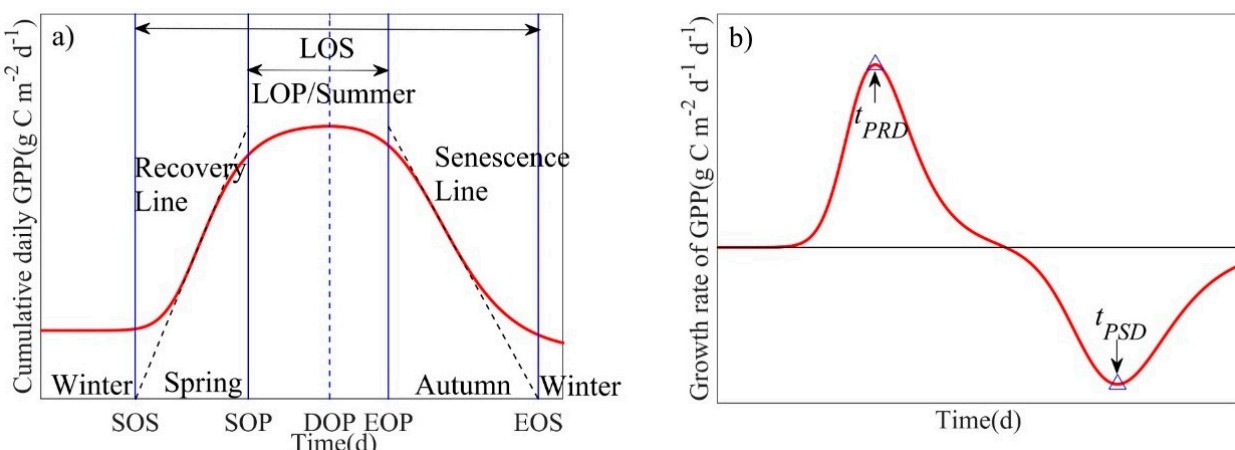

**Figure 2.** Schematic diagram of annual ecosystem-scale phenological process characteristics derived from measured cumulative daily gross primary production (GPP). (**a**) $A(t)$ is the fitted curve shown in red, and the recovery line ($A_{RL}(t)$) and the senescence line ($A_{SL}(t)$) are shown as black dotted asymptotic lines on the left and right, respectively. (**b**) $k(t)$ is the fitted curve shown in red, where $t_{PRD}$ denotes peak recovery day (d), and $t_{PSD}$ denotes peak senescence day (d) [26].

### 2.5. Response of Phenology Model to Disturbance and EOS Correction Approaches

The study area experienced two years of normal rainfall which were followed by 2.5 years of long-term drought (Palmer Drought Severity Indices, PDSI < −2), then 4.5 years of normal or slightly above-average rainfall [36]. In some years, the phenology model indicated that the ecosystem growing season covered the entire year and extended into the next year. In these cases, the end of growing season was truncated to the end of the current year (365 d).

In response to water stress after mid-spring, daily GPP declined rapidly [36]. When the maximum rate of GPP decline during summer (June to August) following this short-term drought was greater than the maximum rate of decline during autumn (September to November), the phenology model underestimated the length of the growing season of the ecosystem, resulting in earlier EOS predictions (Figure 3a). However, this short-term drought did not affect the prediction of the SOS. In these cases, we corrected the EOS via two methods: (1) EOS was predicted based on the second highest maximum rate of decline in GPP in autumn using the phenology model itself ($A_{SL}$), i.e., we used the second highest

maximum rate of decline in GPP as the slope of the $A_{SL}$ to calculate the EOS (Figure 3); or (2) The autumn date at which GPP recovered to the value of GPP at SOS was used as the corrected EOS. Where possible, the first correction method was used; however, when the phenology model yielded unrealistic predictions for the EOS correction (e.g., estimated EOS exceeded the established time scale), the second method was used. Since low-intensity prescribed fires occurred during the dormant season between winter and early spring (January–March) [37,45], the phenology model should capture the potential impacts of fire on SOS.

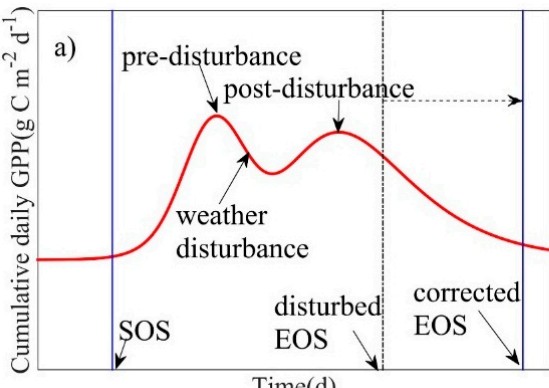 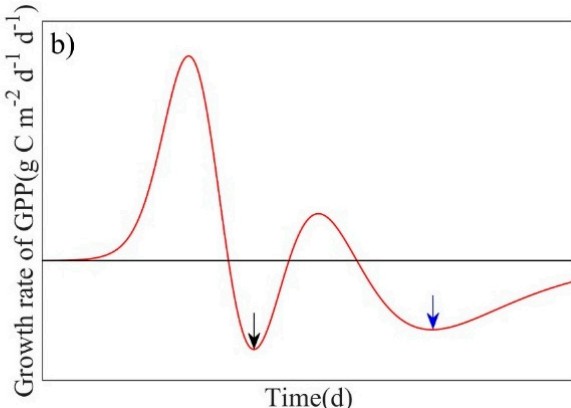

**Figure 3.** Schematic diagram of the influence of weather events on phenological process characteristics at ecosystem-scale: (**a**) $A(t)$ is shown as the fitted red curve, the black dotted line is the default end of growing season (EOS) output of the phenology model using the first maximum GPP decline rate caused by short-term summer drought as the slope of $A_{SL}(t)$ to predict EOS, the solid blue line on the right is the corrected EOS using the second maximum GPP decline rate as the slope of $A_{SL}(t)$ to predict EOS; (**b**) $k(t)$ is shown as the fitted red curve, the black arrow points to the first maximum GPP decline rate caused by weather disturbance, the blue arrow points to the second maximum GPP decline rate after weather disturbance in autumn.

### 2.6. Identifying Anomalous Ecosystem Response to Weather Events Using GPP-Derived Phenological Process

In years without weather disturbances, the modeled cumulative daily GPP followed a bell-shaped pattern whereby the curve peaked between the SOP and EOP (Figure 4a). During the study, however, site-level water stress (short-term droughts, longer-term regional droughts, and heavy rain resulting in flooding) occurred in some years [38], which altered this pattern. To discuss the effects of these weather disturbances, we used two methods to identify whether the ecosystem had an anomalous response from the perspective of the estimated phenology model: (1) when LOP (Figure 2) of the GPP-derived phenological process was estimated at less than 60 d (Figure 4a) [46], (2) when multiple maximum and minimum peaks were observed (Figure 4c) in the GPP growth rate (Figure 4b).

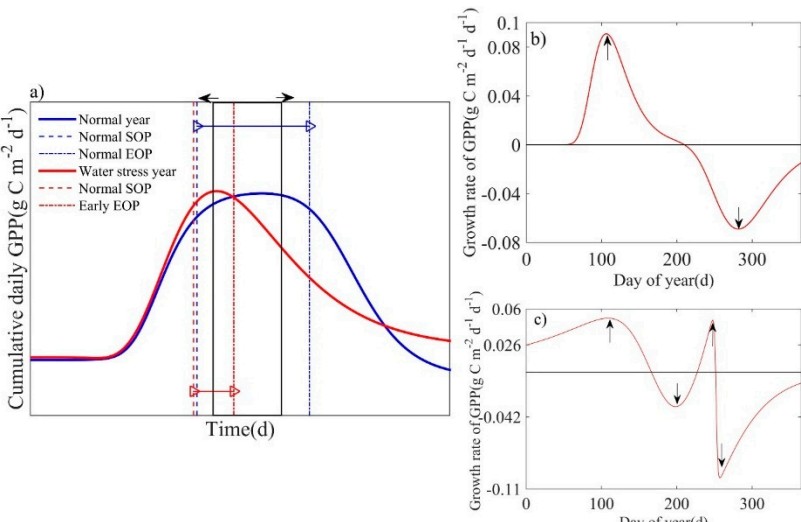

**Figure 4.** Schematic diagram of: (**a**) pattern of modeled cumulative daily GPP, solid black rectangle represents the standard length of peak (LOP) threshold (60 d), blue double-headed arrow indicates the normal LOP (>60 d), and the red double-headed arrow indicates the water-limitation induced short LOP (<60 d); and characteristic GPP growth rate in (**b**) normal year (2013) (black arrow point at the local extrema), with one maximum and minimum peak value (0.09 and −0.06 g C m$^{-2}$ d$^{-1}$ d$^{-1}$, respectively), and (**c**) short-term, summer drought year (2010), with multiple maximum and minimum peak values.

### 2.7. Response of Summer Day-to-Day Phenological Process

Previous studies have reported the long-term precipitation dynamics of these sites during the study period (2009–2017) [40,42], and showed short-term seasonal annual drought and water stress from June to August [38]. This caused significant day-to-day fluctuations in summer ecosystem productivity (GPP decline and recovery) and changes in summer phenological processes across sites.

To study the impact of weather events on summer phenology, here we defined the length of response to stabilization (LORS, d) as the number days between the start of disturbance (SOD) and peak of recovery (POR) (Figure 5). The length of fading (LOF, d) was defined as the number of days between SOD and ROD, and the length of recovery (LOR, d) as the number of days between ROD and POR. The fading rate (g C m$^{-2}$ d$^{-1}$ d$^{-1}$) was defined as the slope of GPP fading between SOD and ROD, and the recovery rate (g C m$^{-2}$ d$^{-1}$ d$^{-1}$) as the slope of GPP recovery between ROD and POR.

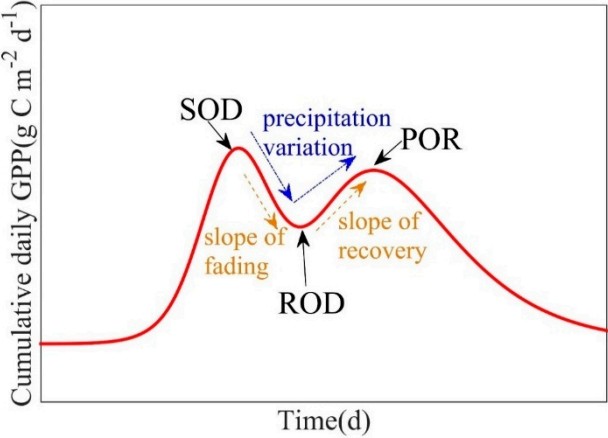

**Figure 5.** Schematic diagram of summer phenological process after short-term, summer drought. SOD = start of disturbance (d), when modeled GPP starts to decline after the spring peak; POR = peak of recovery (d), when modeled GPP reaches peak recovery; ROD = recovery day of disturbance (d).

### 2.8. Long-Term Climate Data from NOAA

In order to study the sensitivity of the phenological processes to changes in air temperature (°C) and precipitation (mm) at monthly and inter-annual scale, we obtained long-term air temperature and precipitation data (1939–2000) from NOAA's National Centers for Environmental Information (NCEI) (station ID: CAMILLA 3 SE, GA US; [47]), using it as a baseline for the calculation of air temperature and precipitation anomalies for the study landscape (Appendix A, Figure A1).

### 2.9. Statistical Analysis

The fit of the phenology model was verified by examining the adjusted coefficient of determination ($R^2$). Analyses were then formulated to test hypotheses using the phenology model parameters. SOS, EOS, and LOS derived from cumulative daily GPP were compared by site using multivariate analysis of variance (MANOVA). Site differences in the multivariate response of these parameter estimates were evaluated with Pillai's Trace, Wilks' Lambda, Hotelling's Trace, and Roy's Largest Root. The *p*-value, *F*-value and $R^2$ were used to evaluate the strength of the univariate relationships between each of these phenological parameters by site [48]. As all sites had prescribed fire applied during the dormant season in 2009, 2011, 2013, 2015 and 2017, we also compared the SOS, EOS, LOS, $t_{PRD}$ and $k_{PRR}$ in fire years versus non-fire years across the three sites.

To evaluate the weather system as a potential phenology driver, we computed the Spearman correlation coefficient to quantify the effects of air temperature, precipitation, and radiation (PAR, $\mu$mol m$^{-2}$ s$^{-1}$) anomalies on ecosystem-scale phenological processes by modeling SOS, EOS, and LOS, at inter-annual scale by site. We also examined relationships between SOS, EOS, and LOS and values of these climate variables in the month, 3 months prior to spring and months of late autumn (March and January–March, October to November, respectively) [46]. We evaluated the effects of air temperature and precipitation anomalies on SOP and POR using ANCOVA by site at a monthly scale. Model input values of air temperature and PAR were average values, and precipitation was input as a cumulative value. In order to understand the mechanisms driving phenological processes of the study landscape, we analyzed the relationship between SOS and EOS with LOS by site using linear regression analysis. This illuminated the impact of spring phenology and autumn phenology date on the LOS of the study sites. Statistical analyses and visualization were processed in OriginPro 9.1 (OriginLab Corporation, Northampton, MA, USA) and SPSS statistics version 17.0 (SPSS Inc., Chicago, IL, USA). Model assumptions of normality and homoscedasticity were visually evaluated.

## 3. Results

### 3.1. Application of Phenology Model for the Three EC Sites

The $R^2$ indicated a good fit for the phenology model in all years and sites ($R^2 \geq 0.8$). The estimated parameters of the phenology model indicated that GPP differed by year and site, and that sites differed in their response to weather events (Table 2; Appendix B, Figure A2). Although the EC sites are in close proximity to one another and share a similar climatic environment [42], there were still slight differences among sites in terms of phenological characteristics derived from daily GPP. Except for 2012 and 2016, phenological responses of GPP to weather events in the sites were consistent. These years with anomalies corresponded to soil moisture-related water limitations [49], and uneven precipitation in spring [49] (non-drought-related irregular rainfall) (Appendix C, Figure A3).

**Table 2.** Description of the application of the phenology model for the three EC sites at the JCI, Newton, GA, USA.

| Year/Site | Mesic | Intermediate | Xeric |
|-----------|-------|--------------|-------|
| 2009 * | Normal | Normal | Normal |
| 2010 | AR (Water stress) | AR (Water stress) | AR (Water stress) |
| 2011 * | AR (Water stress) | AR (Water stress) | AR (Water stress) |
| 2012 | AR (Uneven spring precipitation) | Normal | Normal |
| 2013 * | Normal | Normal | Normal |
| 2014 | AR (Flood) | AR (Flood) | AR (Flood) |
| 2015 * | AR (Water stress) | AR (Water stress) | AR (Water stress) |
| 2016 | Normal | Normal | AR (Water stress) |
| 2017 * | Normal | Normal | Normal |

Anomalous response: AR; * fire year.

### 3.2. Ecosystem-Scale Phenological Characteristics

Prescribed fire and weather events (meteorological drought and flood) affected ecosystem-scale phenological processes and productivity [40,49], especially summer phenology (Table 2; Appendix B, Figure A2) because of the variability of water availability. To develop a more thorough understanding of the phenological patterns of the study sites, we investigated SOS, EOS and LOS patterns and characteristics at an inter-annual scale (Figure 6; Appendix B, Table A1 and Figure A2). Although the three sites had different soil water availability [42], multivariate tests indicated there were no significant differences in SOS, EOS, and LOS by site ($p > 0.12$) (Appendix B, Table A2). However, the mesic site had the longest average LOS (320 d), and the xeric site had the shortest average LOS (300 d) (Appendix B, Table A1).

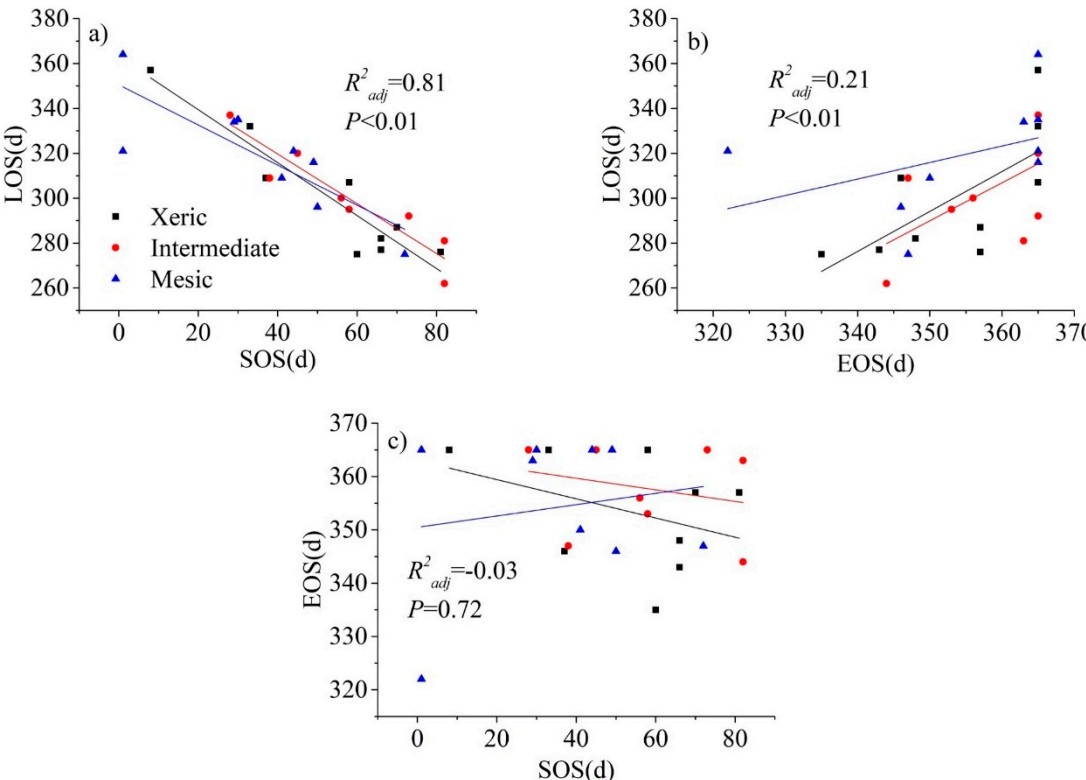

**Figure 6.** Relationship between (**a**) LOS and SOS, (**b**) LOS and EOS, and (**c**) EOS and SOS by site (N = 27). SOS = start of growing season, EOS = end of growing season, LOS = length of growing season.

The relationship between SOS and LOS at all sites was significant and negative (Figure 6; $p < 0.01$); earlier onset of SOS was associated with a significant extension in LOS. The relationship between EOS and LOS was significant and positive for all sites ($p < 0.01$), with a delay of EOS associated with a significant extension in LOS. On the other hand, there was no significant relationship between SOS and EOS; that is, SOS and EOS jointly control LOS across the sites, but SOS and EOS were independent of each other, with their own response mechanisms to environmental and human activities.

### 3.3. Response of Ecosystem-Scale Phenological Processes to Forest Management

The SOS in fire years was delayed by 10 d on average compared to non-fire years, and this trend strengthened along the soil moisture gradient with the mesic site being most affected (18 d delay; Appendix B, Table A1). However, no significant differences were indicated by multivariate tests ($p > 0.14$) (Appendix B, Table A3). Dynamic changes of EOS and LOS showed a weaker relationship with fire; prescribed fire had no discernible effect on the GPP recovery rate of the ecosystem in mid-spring (Appendix B, Table A3; Appendix B, Figure A2), Thus, phenological processes of the ecosystem with higher water availability were more sensitive to prescribed fire, but the responses of EOS and LOS were more subtle than SOS (Appendix B, Table A1).

Furthermore, in years with prescribed fire, SOS values in our study sites were significantly related to PAR and air temperature in the early spring (January to March) ($p < 0.01$; Table 3); as air temperature and PAR increased in spring after fire, the SOS was significantly advanced and the growing season (LOS) was significantly extended ($p < 0.05$). In non-fire years, we did not find similar spring phenology trends ($p > 0.05$; Table 3).

**Table 3.** Spearman correlation coefficients relating start of growing season (SOS), end of growing season (EOS) and LOS (length of growing season) to interannual, March, and early spring (January to March) air temperature ($T_a$), PAR and cumulative precipitation anomalies by site. $T_a$ (°C) and PAR ($\mu$mol m$^{-2}$ s$^{-1}$) were estimated as averages, precipitation as cumulative, for years with and without prescribed fire.

| | | Fire Year | | | Non-Fire Year | | |
|---|---|---|---|---|---|---|---|
| **Time Scale** | **Climate Variable** | **SOS** | **EOS** | **LOS** | **SOS** | **EOS** | **LOS** |
| Annual | $T_a$ anomaly | −0.33 | 0.05 | 0.25 | 0.06 | 0.35 | 0.04 |
| | Precipitation anomaly | 0.31 | 0.11 | −0.23 | −0.11 | −0.30 | 0.04 |
| | PAR anomaly | −0.30 | 0.05 | 0.28 | −0.05 | −0.34 | −0.13 |
| March | PAR | **−0.62 *** | 0.36 | **0.59 *** | −0.29 | −0.17 | 0.13 |
| | $T_a$ | 0.09 | −0.21 | −0.15 | −0.07 | **0.51 **** | 0.28 |
| | Precipitation | 0.39 | −0.13 | −0.35 | 0.10 | −0.17 | −0.08 |
| Early spring | PAR | −0.38 | 0.3 | 0.4 | −0.11 | −0.34 | −0.08 |
| | $T_a$ | **−0.50 *** | **0.56 **** | **0.57 **** | −0.10 | **0.53 **** | 0.32 |
| | Precipitation | 0.06 | 0.21 | 0.02 | −0.17 | −0.47 | −0.01 |

Bold fonts represent significant relationships, * $p < 0.05$; ** $p < 0.01$.

### 3.4. Inter-Annual Climate Control Factors for Ecosystem-Scale Phenological Processes

Ecosystem-scale phenological processes (SOS, EOS, and LOS) showed different sensitivities to annual weather variables by site (Table 4). At the mesic site, SOS was sensitive to PAR in early spring (January to March) and showed a significant negative correlation, i.e., with the increase in PAR in early spring, SOS was significantly advanced ($p < 0.01$), and then the growing season was significantly extended ($p < 0.05$). EOS has a significant positive correlation with annual precipitation ($p < 0.05$), i.e., as the annual precipitation increases, EOS appears to be significantly delayed.

**Table 4.** Spearman correlation coefficients relating start of growing season (SOS), end of growing season (EOS) and LOS (length of growing season) to interannual, March, early spring (January to March), and late autumn (October to November) air temperature ($T_a$), PAR and cumulative precipitation (Pptn) anomalies (mm) by site. $T_a$ (°C) and PAR ($\mu$mol m$^{-2}$ s$^{-1}$) were estimated as averages, precipitation as cumulative.

| Time Scale | Climate Variables | Mesic | | | Xeric | | | Intermediate | | |
|---|---|---|---|---|---|---|---|---|---|---|
| | | SOS | EOS | LOS | SOS | EOS | LOS | SOS | EOS | LOS |
| Annual | $T_a$ anomaly | 0.01 | −0.04 | −0.02 | **−0.70 *** | **0.69 *** | **0.90 *** | −0.15 | 0.25 | 0.08 |
| | Pptn anomaly | 0.14 | **0.64 *** | 0.14 | **0.55 *** | −0.33 | −0.28 | **0.76 *** | −0.34 | **−0.71 *** |
| | PAR anomaly | −0.42 | −0.37 | 0.32 | −0.49 | 0.00 | 0.02 | −0.19 | −0.07 | 0.19 |
| March | PAR | **−0.63 *** | −0.19 | 0.40 | −0.46 | 0.22 | 0.20 | **−0.57 *** | 0.34 | **0.63 *** |
| | $T_a$ | 0.03 | −0.19 | −0.17 | −0.49 | **0.54 *** | **0.75 *** | −0.12 | 0.22 | 0.04 |
| | Pptn | 0.10 | 0.33 | 0.03 | 0.32 | −0.21 | −0.30 | 0.50 | −0.10 | −0.46 |
| Early spring | PAR | **−0.59 *** | −0.13 | **0.50 *** | −0.41 | 0.30 | 0.03 | −0.17 | 0.13 | 0.22 |
| | $T_a$ | −0.17 | 0.44 | 0.22 | −0.46 | **0.69 *** | **0.68 *** | −0.35 | **0.51 *** | 0.33 |
| | Pptn | −0.04 | 0.44 | 0.26 | 0.44 | **−0.52 *** | **−0.57 *** | 0.31 | −0.16 | −0.28 |
| Late autumn | PAR | −0.50 | −0.28 | 0.41 | −0.20 | −0.41 | −0.32 | −0.21 | 0.17 | 0.26 |
| | $T_a$ | −0.15 | −0.24 | 0.06 | −0.10 | −0.04 | 0.20 | 0.32 | 0.19 | −0.31 |
| | Pptn | 0.48 | 0.14 | −0.23 | **0.87 *** | −0.28 | **−0.65 *** | 0.50 | **−0.74 *** | **−0.59 *** |

Bold fonts represent significant relationships, * $p < 0.05$; ** $p < 0.01$. Pptn, precipitation.

At the intermediate site, SOS was sensitive to annual precipitation and spring PAR (March), i.e., as annual precipitation increased, SOS was significantly delayed ($p < 0.05$), but as early spring PAR increased, SOS was significantly advanced ($p < 0.01$). EOS was sensitive to the precipitation at late autumn (October to November). As the precipitation at late autumn increased, EOS was advanced. LOS was also sensitive to precipitation (annual and late autumn) and spring PAR. As precipitation increased, LOS was shortened ($p < 0.01$), and as spring PAR increased, LOS was extended ($p < 0.01$).

At the xeric site, SOS, EOS, and LOS were all sensitive to the air temperature, i.e., as the air temperature warmed, SOS was significantly advanced and EOS was significantly delayed ($p < 0.01$), thereby significantly extending the growing season ($p < 0.05$). However, similar to the intermediate site, precipitation had a significant positive correlation with SOS ($p < 0.05$) and a significant negative correlation with LOS ($p < 0.01$), i.e., as annual and late autumn precipitation increased, SOS was significantly delayed and EOS was significantly advanced, thus significantly shorten the growing season ($p < 0.05$).

In general, we believed that the phenology process of mesic and intermediate sites was more sensitive to PAR in spring, while the xeric site was more controlled by air temperature. More annual precipitation was beneficial to the mesic site, but excess precipitation in late autumn may have a negative impact on xeric and intermediate sites.

*3.5. Response of Ecosystem-Scale Phenological Processes to Weather Disturbances*

All study sites experienced a severe, but short-term drought in March 2017; although there were no extreme air temperature events during that month (16.5, 16.7 and 16.7 °C, respectively), precipitation was 12.95, 13.97 and 11.17 mm mo$^{-1}$ for mesic, intermediate, and xeric sites, respectively (Appendix D, Figure A4). This resulted in anomalously high values of SOP during this year at all sites (Figure 7). Excluding this short, severe drought event, we found that as March air temperature anomalies increased, the SOP was significantly advanced in the mesic and intermediate sites, but there was no similar pattern in the xeric site. Due to this 2017 drought event, the SOP was delayed by 176, 148, and 162 d, at the mesic, intermediate, and xeric sites, respectively (Figure 7).

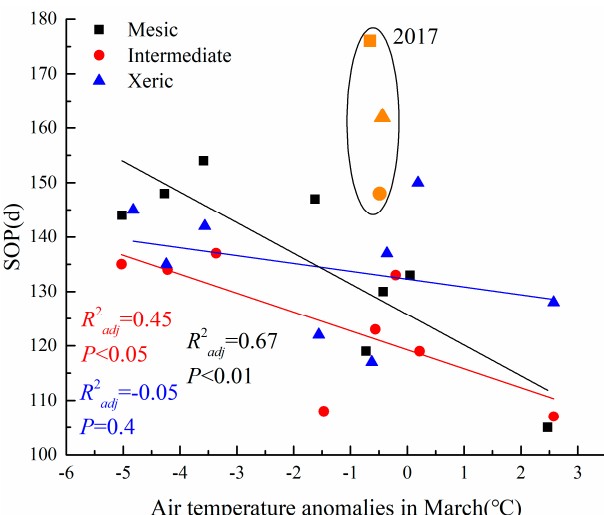

**Figure 7.** Sensitivity of the start of peak (SOP) to March air temperature anomalies from 2009 to 2016 by site.

Based on an analysis of the length of recovery (LOR; Appendix E, Table A4), we found that the ecosystem-scale phenological response during the summer was different by site following short-term drought. The start of disturbance (SOD) was generally located in the early summer and late spring (130–160 d). Timing of recovery from disturbance (ROD) depended on when precipitation started to recover after this short-term water stress, and the peak of recovery (POR) was generally in the end of August (up to 252 d) after the restoration of precipitation. If precipitation did not recover after short-term, summer drought, it caused ecosystem GPP to continue to decline until EOS (autumn) with a short summer (<60 d).

Short-term summer drought lasted from June to August in some years, and in these cases summer GPP did not recover (no POR) (Appendix E, Table A4). POR was significntly correlated with August precipitation (mm; $p < 0.01$). With increased precipitation in August after a short-term drought, POR was significantly delayed (end of summer) across the sites (Appendix F, Figure A5).

## 4. Discussion

This study explored the phenological process of three subtropical longleaf pine forest sites with different soil water-holding capacities over nine years, which included phenological responses to prescribed fire and varying weather events. Our results supported our hypotheses, suggesting that soil water-holding capacity affected the growing season length. Low-intensity prescribed fire did not cause significant physiological stress on plant communities, and thus had a small effect on early spring phenology with short-term shifts of aboveground biomass; however, it may increase the sensitivity of early spring phenology to radiation and air temperature. Dynamics in water availability caused by periodic weather events affected the senescence mechanism of plant communities, and in this process air temperature, precipitation and radiation were found to be the main climatic drivers for vegetation carbon phenology. Restoration of precipitation after a short-term summer drought reduced the lagged effect of water stress (Appendix F, Figure A5), resulting in a positive phenological response in ecosystem productivity to precipitation restoration after short-term water stress. In addition, we found evidence of the adaptability of plant community photosynthesis to long-term precipitation dynamics which affect ecosystem phenological response (Figure 7) (Appendix A, Figure A1).

### 4.1. Impact of Climate Change on the Phenological Process under Short-Term Weather Events

Forests have a variety of environmental drivers that are known to control their growing season length [20,30], and therefore their physiological functions. In longleaf pine

ecosystems of the southern United States, growing season is influenced by weather anomalies such as flood and drought, which interact with the system's climate (Appendix B, Figure A2). Air temperature was one of the main drivers of phenological variation during our study period (Table 4) [2], in agreement with Li et al. [50]. This was also consistent with global warming trends that are shown to control the phenology of temperate forests by contributing to earlier onset of SOS with warmer conditions [32]. On the other hand, due to the differences in soil water holding capacity at the site-level, excessive precipitation may shorten the growing season due to delayed SOS and advancement of EOS (Table 4) [51]. However, for the mesic site, precipitation aided in maintaining higher soil water content which may have delayed vegetation senescence in autumn (Table 4). Periodic weather events also appear to contribute to longleaf ecosystem phenological responses in summer (Appendix B, Figure A2; Appendix F, Figure A5) [49].

### 4.2. Effects of Soil Water Availability Gradients on Length of Growing Season at the Site-Level

Soil water availability is an important driver that affects plant growth and photosynthetic capacity [52,53], and may be a potential phenological control in this study [54]. However, due to the 100-year average tree age and co-evolution of the vegetation [42], canopy and understory communities may have adapted to soil water availability [31], resulting in a similar phenological pattern by site with differences in the specific dates of phenology events. This means that in relatively warm subtropical regions, where radiation and air temperature are at higher levels, greater water availability could prolong LOS (Appendix B, Table A1) and may be one of the primary control factors of productivity in forest ecosystems [55].

### 4.3. Impact of Prescribed Fire on Ecosystem-Scale Phenology in Early Spring

Prescribed fire is a common forest management method in the southern United States [56] and has been shown to delay the timing of understory green-up (Appendix B, Table A1). Since prescribed fire is implemented during the dormancy period (winter to early spring) [44,45], burning of the understory caused short-term shifts in aboveground biomass and photosynthetic capacity [57]. Fire may increase soil nutrients, which can promote rapid recovery of herbaceous plants and thus ecosystem productivity [56,58]. This indirectly improves light use efficiency, which may increase the sensitivity of spring phenology to air temperature and PAR (Table 3). This may have contributed to the slight difference in SOS over the course of the study, since low-intensity prescribed fires did not cause functional disruption of forest structure and physiological function [44,45,58].

Site-level soil water holding capacity was an important influencing factor for the sites' phenological response to prescribed fire. Increased post-fire soil hydrophobicity has been shown to lead to short-term water competition, which limits the development of understory leaves, thus reducing GPP which drives our phenological model [59]. Greater soil water holding capacity is related to greater understory plant diversity and biomass [40], which resulted in a relatively longer recovery time for productivity following fire. To fully understand the meteorological and biological mechanisms of how prescribed fire influences spring phenology, it may be necessary to study fire intensity by site, as greater fire intensity may be associated with more vegetation loss on ignition, leading to a longer period of recovery [59].

### 4.4. Adaptability of Plant Community Photosynthesis to Spring Precipitation

Warmer air temperature in spring can increase the rate of leaf out for the understory, which in turn can advance plant photosynthesis and may lead to greater carbon sequestration [32]. Under the background of seasonal spring warming, if there was no severe drought during the green-up phase, the photosynthetic capacity of the plant communities was unaffected, advancing the start of summer (SOP). One potential reason for this phenomenon may be the long-term regional climate; March is historically the second highest month of precipitation (Appendix A, Figure A1), and the plant community has already

adapted to higher precipitation during this time, which allows for accelerated green-up. However, when short-term drought occurs in March, it delays the green-up phase, reducing productivity and delaying SOP.

### 4.5. Response of Ecosystem-Scale Summer Phenology

Since the three study sites are in a warm subtropical region, rarely are there light or temperature limitations which constrain GPP. Therefore, water availability is an important factor which supports and affects GPP (understory and canopy; Appendix F, Figure A5) and thus day-to-day phenological processes at the ecosystem scale [40] (Appendix B, Figure A2).

Water availability has been shown to have a lagged effect on productivity of subtropical forests [50]. Our study weakly supports this finding with evidence in the intermediate site in 2010 when precipitation led to a small increase in GPP after short-term drought, but this recovery was not significant compared to pre-drought physiology. The reason may be that after the drought event in July (34.3 mm mo$^{-1}$), the recovery of precipitation in August (85.3 mm mo$^{-1}$) was not enough to offset the water deficit which occurred the month prior (Appendix B, Figure A2). Subsequent above average precipitation is required to offset carbon loss during water stress (Appendix A, Figure A1) and stabilize photosynthetic capacity in summer, as additional precipitation demand is related to evapotranspiration during and after water stress [60]. In addition, increased precipitation following the drought offset carbon loss by reaching the "effective precipitation" threshold, which is defined as the ratio of precipitation to evaporation [61]. This increased water availability had a positive effect on GPP (Appendix F, Figure A5). However, when short-term drought lasted from June to August, the ecosystem EOP date was advanced which led to a shorter summer at all sites (Appendix B, Figure A2). This may have been associated with the fact that many conifers—when experiencing drought—will flush one cohort of needles earlier in the summer to limit water loss and reduce drought stress which will reduce GPP rates [62].

We also observed that following summer water stresses, August precipitation shortened the senescence phase (Appendix F, Figure A5). While September precipitation had less of an effect on the end date of summer, this corresponded more closely to a decline in photoperiod and a reduction in air temperature [15] (Appendix A, Figure A1).

Although sufficient precipitation during the growing season can extend the end of summer (Appendix F, Figure A5), precipitation is not the only controlling factor for summer ecosystem productivity. Higher vapor pressure deficit (VPD) is known to cause stomatal limitation in longleaf pine ecosystems, which leads to declines in GPP [44,45,63]. The sites experienced this type of event in the summer of 2015; although drought was absent, there was still a significant decline in GPP associated with higher VPD (Appendix B, Figure A2). Yet we as a scientific community still have not determined the specific tipping point when stomatal limitations would limit GPP in this system. Thus, to better understand the effect of summer phenological processes of subtropical forests associated with VPD, additional experiments are needed that link physiological activity to VPD thresholds.

### 4.6. Management Implications

Prescribed fire is a widespread practice in the southeastern region of the United States, especially on lands where restoration is a primary management goal [35]. Fire resulted in moderate delays of the SOS, which were stronger in the site that had higher water availability; however, fire did not impact other subsequent phenology parameters. Our results also suggested a slightly amplified association between SOS and weather variables, such as radiation and air temperature, during years of prescribed fire. However, these impacts may be mitigated by projected changes in climate, which include global warming trends associated with earlier onset of SOS [32]. Perhaps a larger concern for land managers is the interaction of prescribed fire with drought conditions, which may advance the EOS and also become more common under a future climate [35]. However, a longer time series is needed to quantify these interactions and rigorously address this question, as well as study

other management techniques, such as thinning, given underlying natural variability in climatic conditions and the expectation that this variance will increase with climate change.

### 4.7. Limitations and Challenges

We found that the use of phenological models in describing growing season characteristics with daily GPP still leaves uncertainty in our understanding of longleaf pine ecosystems; during short-term summer drought and post-drought recovery, the phenological model may not be able to appropriately reflect dynamic changes with GPP which reduces performance [48]. Improving the prediction accuracy of phenology models and developing piecewise logistic functions to deal with natural and human induced disturbances are still areas of critical research need that need to be examined in future studies. This will aid in advancing our predictive capacity of key forest phenological periods [24].

Discussing the impact of weather and forest management on ecosystem phenological processes still requires a longer series of environmental observations. This study, while encompassing nine years, shows only a subset of possible conditions experienced in this site. Finally, the study is only a preliminary discussion of the phenological processes of subtropical evergreen coniferous forest ecosystems under human and weather disturbances and their responses to disturbances and climate changes. To analyze the influence of a single climatic factor and the threshold of the ecosystem's demand for precipitation after short-term drought, more environmental observations and physiological experiments are needed.

## 5. Conclusions

Using a total of 27 site-years of GPP and environmental observation data, the phenological characteristics of three subtropical forest sites with different soil water availability were discussed. We found that the impact of prescribed fire on ecosystem-scale phenological processes is limited. In this study, fire only weakly affected the timing of the start of the growing season and had no direct impact on subsequent ecosystem phenological processes. Increased air temperature and spring radiation were found to significantly extend the growing season. Increased annual precipitation positively impacted sites with better soil water holding capacity; however, excessive late autumn and winter precipitation may have a negative impact on early spring and autumn phenology in sites with poor soil water holding capacity. Restoration of precipitation after short-term summer drought is extremely important for the recovery of ecosystem productivity, which will directly affect the end of summer, and sufficient precipitation in August will extend the end of summer. This is of particular importance for subtropical forests, as the timing of precipitation is predicted to be altered with climate change. Further study of the phenology of these forests could greatly improve our understanding of future GPP patterns, and ultimately their carbon dynamics.

**Author Contributions:** Conceptualization, Y.G. and C.L.S.; methodology, Y.G.; software, Y.G.; formal analysis, Y.G.; investigation, G.S.; data curation, S.W.; writing—original draft preparation, Y.G., G.S. and C.L.S.; writing—review and editing, Y.G., Y.Z., C.L.S., S.W. and G.S.; supervision, G.S. and C.L.S.; project administration, G.S.; funding acquisition, G.S. All authors have read and agreed to the published version of the manuscript.

**Funding:** Y.G. was funded by the Priority Academic Program Development of Jiangsu Higher Education Institutions (PAPD). C.L.S. and G.S. acknowledge support from the U.S. National Science Foundation (DEB EF-1241881). C.L.S. also acknowledges support from the U.S. National Science Foundation (DEB EF-1702029).

**Institutional Review Board Statement:** Not applicable.

**Informed Consent Statement:** Not applicable.

**Data Availability Statement:** Publicly available datasets were analyzed in this study. This data can be found here: (https://ameriflux.lbl.gov/) as sites US-LL1, US-LL2, and US-LL3.

**Acknowledgments:** The authors thank the members from the Starr laboratory (The University of Alabama) for their constructive comments on the revision of the article. The authors also thank the National Key Research and Development Program of China (No. 2016YFC0502704) for financial support.

**Conflicts of Interest:** The authors declare no conflict of interest.

## Appendix A

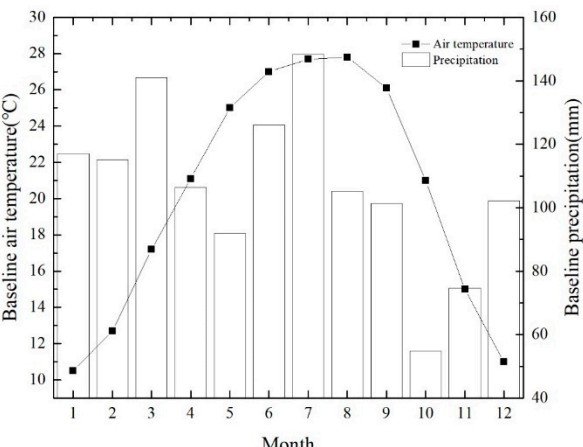

**Figure A1.** Long-term air temperature and precipitation baseline for the three EC flux sites from 1939 to 2000. The baseline annual average air temperature and cumulative precipitation for the three study EC sites were 20.17 °C and 1283.82 mm [47].

## Appendix B

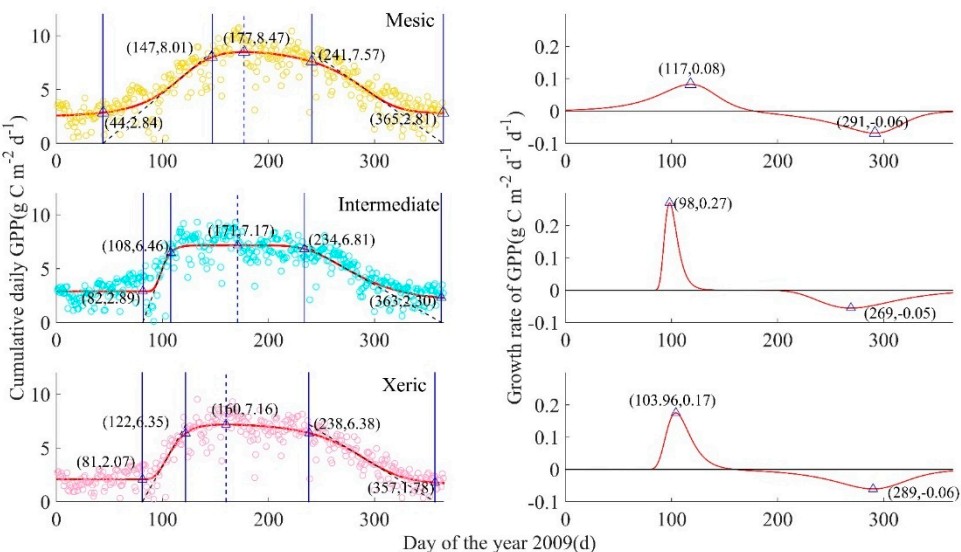

**Figure A2.** *Cont.*

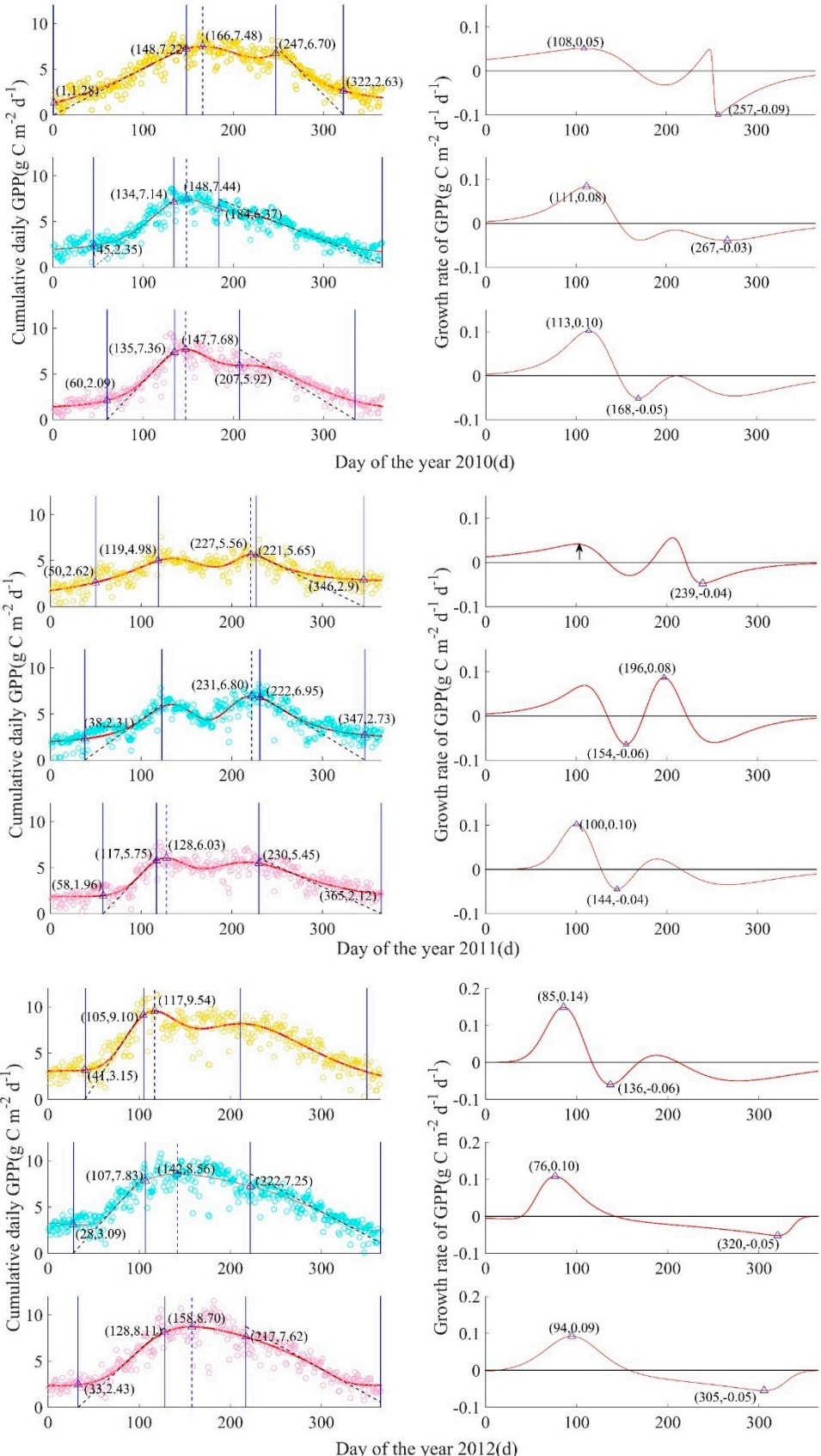

**Figure A2.** *Cont.*

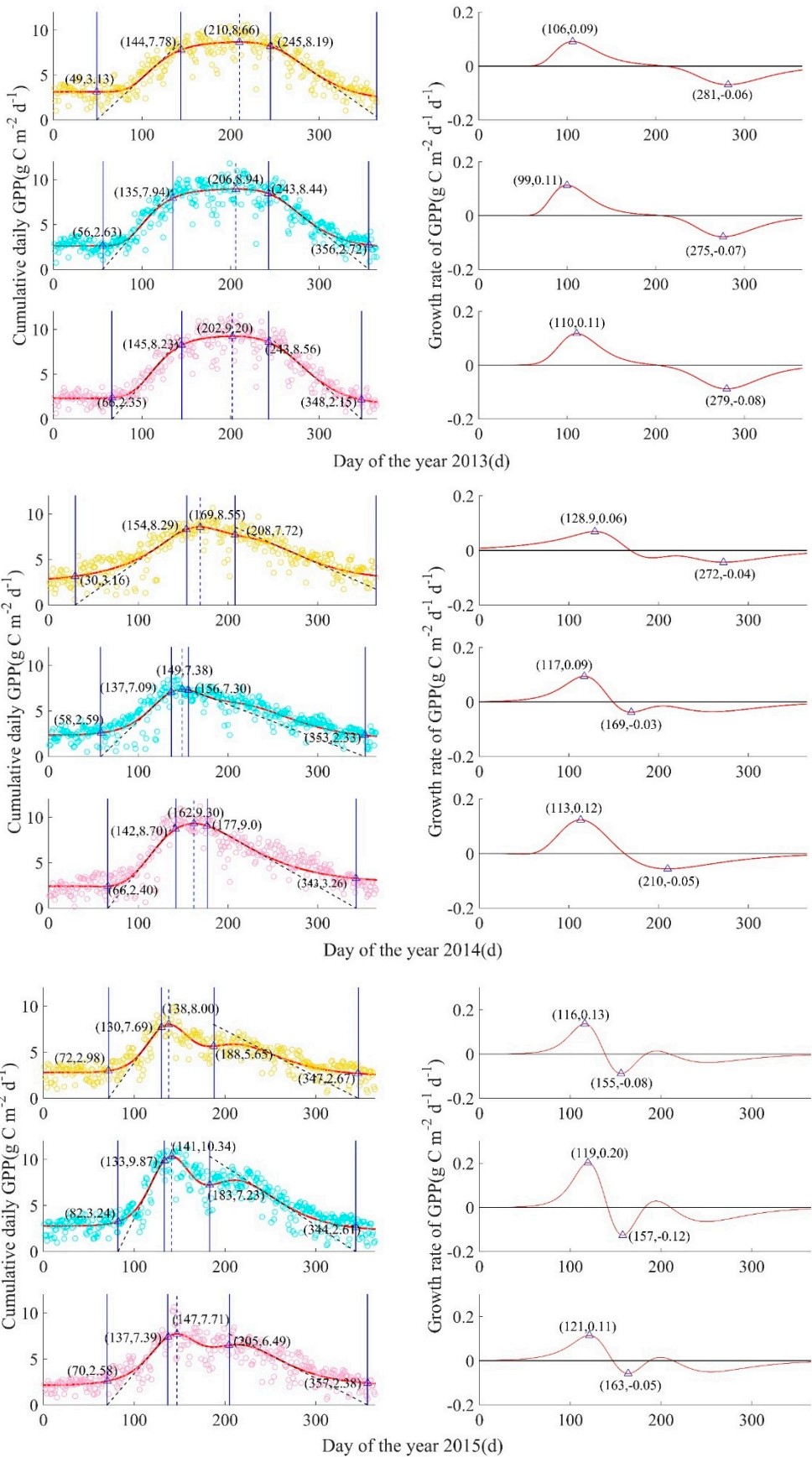

**Figure A2.** *Cont.*

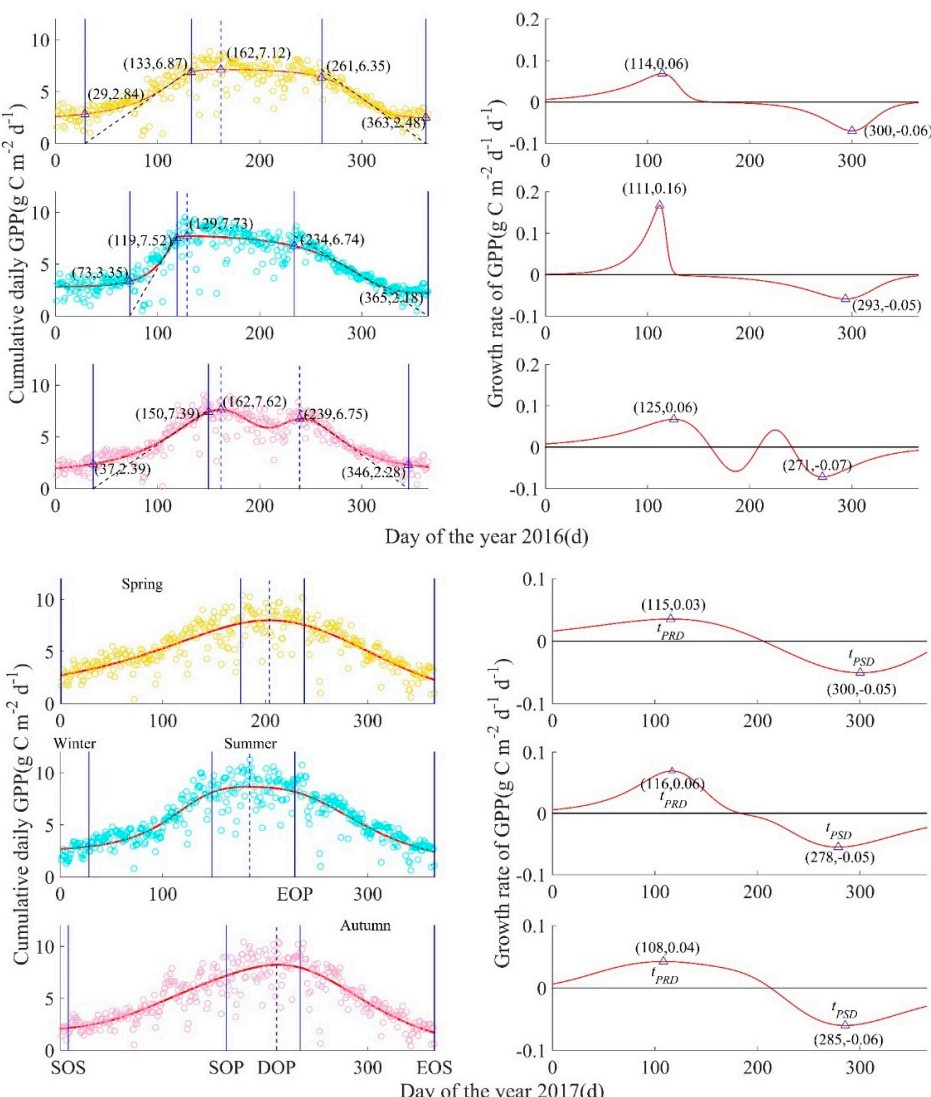

**Figure A2.** GPP-derived ecosystem-scale phenological process for mesic, intermediate, and xeric sites by year, from 2009 to 2017. Left column: Circles are measured cumulative daily GPP(g C m$^{-2}$ d$^{-1}$), with mesic in yellow (top), intermediate in turquoise (middle), and xeric in pink (bottom), solid curves are fitted plant growth functions $A(t)$, four solid blue lines from left to right are SOS (start of growing season), SOP (start of peak), EOP (end of peak), EOS (end of growing season), and the blue dotted line is at the date of the modeled maximum daily GPP (DOP = day of peak). Right column: Red curves represent the GPP growth rate function for each corresponding site, and triangles represent the peaks of the GPP growth rate function, such that $t_{PRD}$ is the peak recovery date, the maximum recovery point of GPP, and $t_{PSD}$ is the peak senescence date, the maximum senescence point of GPP; the date when the GPP growth rate equals zero was the date of the peak of modeled cumulative daily GPP.

There was no summer short-term drought or extreme air temperature events in the mesic site in 2009 [37] and thus this site-year was representative of phenological characteristics associated with normal weather conditions (Appendix B, Figure A2). The spring recovery line (RL, $A_{RL}(t)$) had a slope of 0.08 g C m$^{-2}$ d$^{-1}$ d$^{-1}$, and the autumn senescence line (SL, $A_{SL}(t)$; right black dotted line) had a slope of −0.06 g C m$^{-2}$ d$^{-1}$ d$^{-1}$. Modeled maximum daily GPP occurred at 177 d (8.47 g C m$^{-2}$ d$^{-1}$), which corresponds with the date when GPP growth rate is zero. The $t_{PRD}$ occurred in mid-spring (117 d, 0.08 g C m$^{-2}$ d$^{-1}$ d$^{-1}$), and $t_{PSD}$ occurred in mid-autumn (291 d, −0.06 g C m$^{-2}$ d$^{-1}$ d$^{-1}$).

During 2009 in the mesic site, summer was estimated to occur from 147 d to 241 d, and autumn was from 241 d to 365 d with a length of 124 d. The ecosystem-scale growing season was from 44 d to 365 d, indicating that the LOS was 321 d. The modeled daily peak photosynthetic capacity was 8.47 g C m$^{-2}$ d$^{-1}$ with a LOP/summer of 94 d. The length

order of four seasons for the mesic site in 2009 was: autumn > spring > summer > winter. This ecosystem remained photosynthetically active for most of the year in 2009, because of the shorter winter (early leaf green-up and late leaf senescence).

**Table A1.** Description of ecosystem-scale phenology of the mesic, intermediate, and xeric sites. SOS = start of growing season, EOS = end of growing season, LOS = length of growing season. F = fire year (odd years), NF = Non-fire year (even years).

| | SOS(d) | | | EOS(d) | | | LOS(d) | | |
|---|---|---|---|---|---|---|---|---|---|
| **Year** | **Mesic** | **Inter.** | **Xeric** | **Mesic** | **Inter.** | **Xeric** | **Mesic** | **Inter.** | **Xeric** |
| 2009 | 44 | 82 | 81 | 365 | 363 | 357 | 321 | 281 | 276 |
| 2010 | 1 | 45 | 60 | 322 | 365 | 335 | 321 | 320 | 275 |
| 2011 | 50 | 38 | 58 | 346 | 347 | 365 | 296 | 309 | 307 |
| 2012 | 41 | 28 | 33 | 350 | 365 | 365 | 309 | 337 | 332 |
| 2013 | 49 | 56 | 66 | 365 | 356 | 348 | 316 | 300 | 282 |
| 2014 | 30 | 58 | 66 | 365 | 353 | 343 | 335 | 295 | 277 |
| 2015 | 72 | 82 | 70 | 347 | 344 | 357 | 275 | 262 | 287 |
| 2016 | 29 | 73 | 37 | 363 | 365 | 346 | 334 | 292 | 309 |
| 2017 | 1 | 28 | 8 | 365 | 365 | 365 | 364 | 337 | 357 |
| AVG of F | 43 | 57 | 56 | 357 | 355 | 358 | 314 | 297 | 301 |
| AVG of NF | 25 | 51 | 49 | 350 | 362 | 347 | 324 | 311 | 298 |

**Table A2.** Multivariate tests of SOS (start of growing season), EOS (end of growing season), and LOS (length of growing season, respectively) by site from MANOVA.

| Effect | | Estimate | F Statistic | NumDF | DenDF | *p*-Value | Partial $\eta^2$ |
|---|---|---|---|---|---|---|---|
| Intercept | Pillai's Trace | 0.999 | 12,472.2 | 2 | 23 | <0.001 | 0.999 |
| | Wilks' Lambda | 0.001 | 12,472.2 | 2 | 23 | <0.001 | 0.999 |
| | Hotelling's Trace | 1084.5 | 12,472.2 | 2 | 23 | <0.001 | 0.999 |
| | Roy's Largest Root | 1084.5 | 12,472.2 | 2 | 23 | <0.001 | 0.999 |
| Site | Pillai's Trace | 0.186 | 1.229 | 4 | 48 | 0.311 | 0.093 |
| | Wilks' Lambda | 0.818 | 1.212 | 4 | 46 | 0.319 | 0.095 |
| | Hotelling's Trace | 0.217 | 1.193 | 4 | 44 | 0.327 | 0.098 |
| | Roy's Largest Root | 0.190 | 2.286 | 2 | 24 | 0.123 | 0.160 |

DenDF = Denominator degrees of freedom; NumDF = Numerator degrees of freedom.

**Table A3.** Multivariate tests of SOS (start of growing season), EOS (end of growing season), LOS (length of growing season), $k_{PRR}$ (maximum GPP growth rate), and $t_{PRD}$ (date of $k_{PRR}$) by site and application of prescribed fire from MANOVA.

| Effect | | Estimate | F Statistic | NumDF | DenDF | *p*-Value | Partial $\eta^2$ |
|---|---|---|---|---|---|---|---|
| Intercept | Pillai's Trace | 0.999 | 6620.6 | 4 | 18 | <0.001 | 0.999 |
| | Wilks' Lambda | 0.001 | 6620.6 | 4 | 18 | <0.001 | 0.999 |
| | Hotelling's Trace | 1471.2 | 6620.6 | 4 | 18 | <0.001 | 0.999 |
| | Roy's Largest Root | 1471.2 | 6620.6 | 4 | 18 | <0.001 | 0.999 |
| Site | Pillai's Trace | 0.353 | 1.017 | 8 | 38 | 0.44 | 0.176 |
| | Wilks' Lambda | 0.676 | 0.975 | 8 | 36 | 0.47 | 0.178 |
| | Hotelling's Trace | 0.438 | 0.931 | 8 | 34 | 0.50 | 0.180 |
| | Roy's Largest Root | 0.297 | 1.412 | 4 | 19 | 0.26 | 0.229 |
| Fire | Pillai's Trace | 0.132 | 0.683 | 4 | 18 | 0.61 | 0.132 |
| | Wilks' Lambda | 0.868 | 0.683 | 4 | 18 | 0.61 | 0.132 |
| | Hotelling's Trace | 0.152 | 0.683 | 4 | 18 | 0.61 | 0.132 |
| | Roy's Largest Root | 0.152 | 0.683 | 4 | 18 | 0.61 | 0.132 |
| Site x Fire | Pillai's Trace | 0.334 | 0.951 | 8 | 38 | 0.48 | 0.167 |
| | Wilks' Lambda | 0.678 | 0.969 | 8 | 36 | 0.47 | 0.176 |
| | Hotelling's Trace | 0.456 | 0.969 | 8 | 34 | 0.47 | 0.186 |
| | Roy's Largest Root | 0.413 | 1.961 | 4 | 19 | 0.14 | 0.292 |

DenDF = Denominator degrees of freedom; NumDF = Numerator degrees of freedom.

## Appendix C

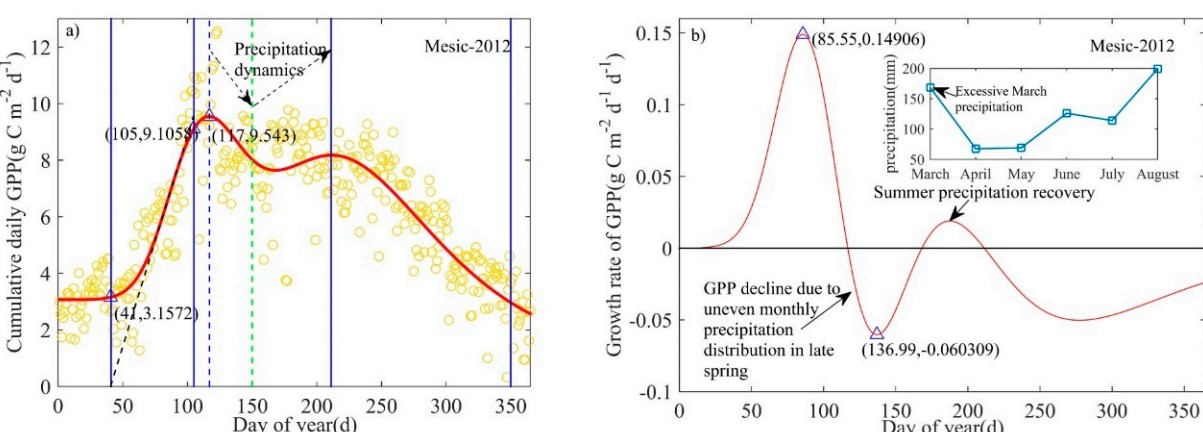

**Figure A3.** Effects of non-drought-related irregular spring rainfall on summer phenological process of mesic site in 2012.

At 117 d in 2012, the mesic site started to have a significant decline in GPP (spring), which was caused by uneven precipitation distribution from March to May [64], resulting in a decline in ecosystem productivity. The restoration of precipitation in June promoted the recovery of ecosystem productivity.

## Appendix D

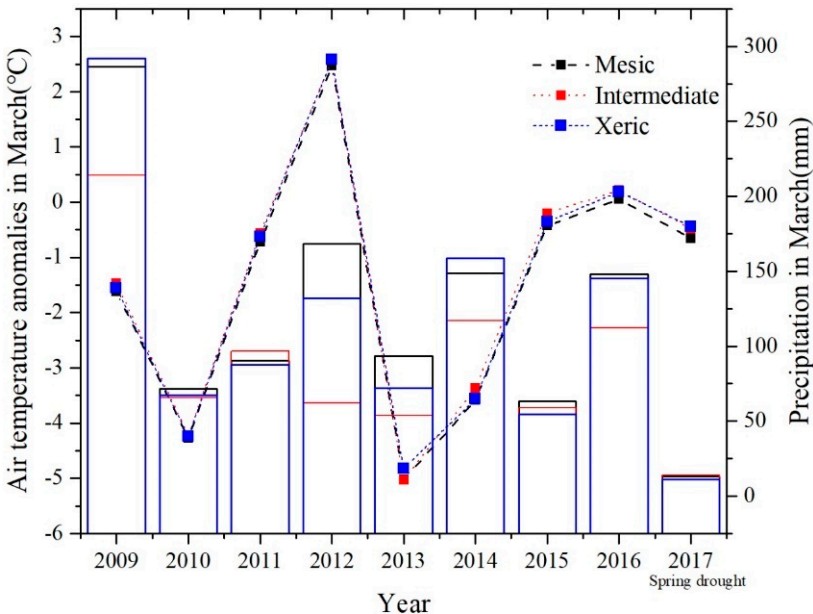

**Figure A4.** Precipitation and air temperature anomaly in March by site (dotted line represents the air temperature, bars represent precipitation).

## Appendix E

**Table A4.** Post-drought ecosystem phenology by site: SOD = start of disturbance (d), ROD = recovery day of disturbance (d), POR = peak of recovery (d), LORS = length of response to stabilization (d), LOF = length of fading (d), LOR = length of recovery (d).

| Site | Year | SOD (GPP) | ROD (GPP) | POR (GPP) | LORS | LOF | LOR | Fading Rate | Recovery Rate |
|---|---|---|---|---|---|---|---|---|---|
| Mesic | 2010 | 167 (7.48) | 227 (6.22) | 252 (6.87) | 85 | 60 | 25 | −0.021 | 0.025 |
| | 2011 | 135 (5.22) | 182 (4.32) | 221 (5.65) | 86 | 47 | 39 | −0.019 | 0.034 |
| | 2014 | 169 (8.55) | NC | NC | NC | NC | NC | NC | NC |
| | 2015 | 138 (8.00) | 184 (5.63) | 210 (5.86) | 72 | 46 | 26 | −0.051 | 0.008 |
| Inter-mediate | 2010 | 148 (7.44) | NC | NC | NC | NC | NC | NC | NC |
| | 2011 | 134 (6.03) | 174 (4.38) | 222 (6.95) | 88 | 40 | 48 | −0.041 | 0.053 |
| | 2014 | 149 (7.38) | NC | NC | NC | NC | NC | NC | NC |
| | 2015 | 142 (10.35) | 183 (7.23) | 211 (7.75) | 69 | 41 | 28 | −0.076 | 0.018 |
| Xeric | 2010 | 147 (7.68) | NC | NC | NC | NC | NC | NC | NC |
| | 2011 | 128 (6.04) | 173 (4.90) | 216 (5.58) | 88 | 45 | 43 | −0.025 | 0.015 |
| | 2014 | 162 (9.30) | NC | NC | NC | NC | NC | NC | NC |
| | 2015 | 147 (7.71) | 188 (6.28) | 214 (6.54) | 67 | 41 | 26 | −0.034 | 0.01 |
| | 2016 | 162 (7.62) | 209 (5.89) | 241 (6.76) | 79 | 47 | 32 | −0.036 | 0.027 |

NC: no recovery, GPP: Gross Primary Production at the day (g C m$^{-2}$ d$^{-1}$). Note: Due to the seasonal summer drought, precipitation did not recover after drought in some site-years, which caused the GPP to decline without recovery (NC) (Figure A2).

On average, LORS was ~2.5 months in length, and slightly higher in the mesic site (81 d) versus the intermediate and xeric site (78 d). The average LOF (40–51 d) was greater than the average LOR (30–38 d), which may indicate that even if precipitation recovered after a short-term drought, the response of GPP recovery to precipitation was delayed.

In the mesic site, the water stress in 2010 occurred in July, and the precipitation was 27.43 mm in that month, but the precipitation returned to normal in August (160.52 mm), which put its POR at early September (252 d). A severe drought occurred in June 2011 with 50.8 mm of precipitation, and precipitation recovered in July (126.23 mm), but the precipitation began to fall again in August (52.32 mm), which brought SOD closer to late spring (135 d), ROD closer to the end of June and POR near mid-August. There was no drought in the summer of 2012 (total precipitation in June-August was 439 mm), and thus the reason for the decline in GPP may be a delayed effect of excessive March rainfall (Appendix B, Figure A2) [64]. Due to the effect of a spring flood event (April, 280.41 mm) and water stress from July to August, there was a lack of recovery in GPP after the end of spring until EOS. A short-term drought occurred in August (30.38 mm) in 2015, but although there was abundant precipitation in June-July, there was still significant GPP decline and the ecosystem entered autumn earlier than normal (210 d). One potential reason could be the dynamics of VPD [63].

For the intermediate site, in 2010, water stress occurred in June-August (total precipitation of 205.48 mm and very low July precipitation of 34.29 mm). Although August precipitation had a weak recovery (85.34 mm), which was reflected in the GPP growth rate (weak recovery trend; Figure A2), it did not offset the carbon loss in July. This led to SOD occurring in the late spring after photosynthesis of the ecosystem reached its peak in the late spring, and GPP continued to decline into autumn until EOS without a recovery. One reason may be that after this severe drought event in July, more August precipitation was needed to recover and stabilize GPP (>85.34 mm), but the threshold of "effective precipitation" after the drought was still uncertain. In 2011, drought occurred in June (51.82 mm) and precipitation recovered in July (115.56 mm) but declined again in August (68.08 mm). This caused ROD to occur at the beginning of July and POR at the middle of August, and the ecosystem entered autumn earlier (222 d) than normal (~240 d). In 2014, Due to the spring flood and water stress in July-August (65.87 and 28.95 mm), GPP continued to decline until EOS without obvious GPP recovery. Water stress occurred in

August (49.27 mm) in 2015, which brought the POR to 211 d; that is, the ecosystem entered autumn early in August until EOS. However, after the spring (140 d), there has been a significant decline in GPP, which is inconsistent with sufficient precipitation from June to July (152.92 and 99.32 mm). One potential reason for that phenomenon may also be the VPD dynamics [63].

In the xeric site in 2010, water stress occurred in June-July (73.41 and 40.89 mm), and precipitation began to recover in August (117.34 mm). Although the recovery of precipitation caused the GPP growth rate to recover, which offset the decline in GPP from June to July, it did not cause a significant GPP recovery. The reasons for this phenomenon are not clear. A severe drought occurred in June and August 2011 (38.86 and 30.22 mm), but July precipitation was normal (160.03 mm). This led to SOD in spring, ROD in late June and early July (188 d), with a POR of 216 d (early start of autumn). Flooding occurred in spring 2014, and the phenology response of the xeric site was consistent with the intermediate and mesic sites. There was no severe drought in the summer of 2015, but GPP decline still occurred, which may have been affected by VPD [63]. In 2016, short-term drought occurred in July (53.84 mm), and precipitation was normal in June and August (105 mm and 145 mm), which caused SOD to occur at the end of June, ROD to occur at the end of July and early August (209 d). Due to normal precipitation in August, ecosystem POR was in normal range (end of August, 241 d).

**Appendix F**

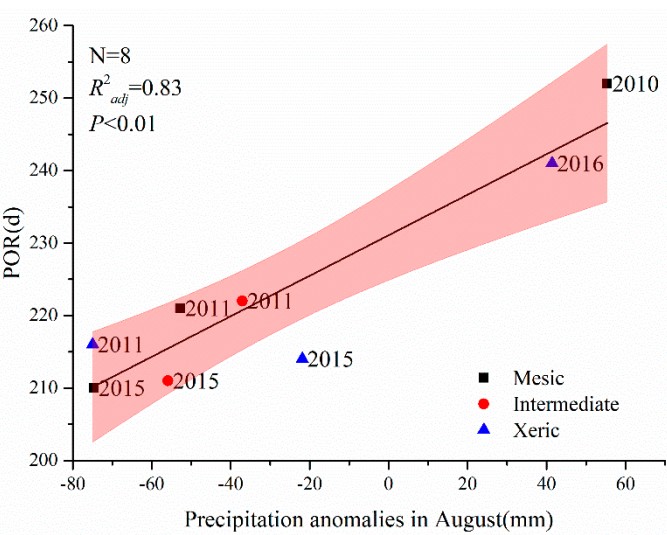

**Figure A5.** Impact of precipitation in August on POR (peak of recovery) after short-term summer drought by site. Pink shaded area represents 95% confidence interval.

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
