# Peer review of "Characterizing Growing Season Length of Subtropical Coniferous Forests with a Phenological Model"

_forests, doi:10.3390/f12010095_

Round 1

Reviewer 1 Report

  1. General comment :

The authors present an interesting phenological study. From my understanding, the technical parts of this work were done carefully and correctly, following an adequate approach. Figures has a good quality, and analyses and writing are adequate. Moreover, I appreciate the effort the authors put in this study and the long-term monitoring.

At this stage, I propose the authors to consider these suggestions in a minor revision. I encourage the authors to incorporate these minor suggestions, and I want to say congratulations for this interesting and original research.

General comments

  • Title is very long and complex. Please, simplify it. E.g remove after “ using…” or replace it for: “: A phenological approach”
  • Keywords could be improved. Some repetitions with the title (subtropical forest ecosystems, phenology model). I suggest add: modelling, ecophysiology, restoration…. The keywords must be presented in alphabetical order.
  • Introduction: An international context is lacking about climate change, and link with prescribed burning must be improved.
  • Table 1: 1 or 2 decimals no both
  • Discussion: Make a new section (small) to talk about management implications.

Author Response

Our replies to Reviewer 1 are below each comment, in italics.

The authors present an interesting phenological study. From my understanding, the technical parts of this work were done carefully and correctly, following an adequate approach. Figures has a good quality, and analyses and writing are adequate. Moreover, I appreciate the effort the authors put in this study and the long-term monitoring.

At this stage, I propose the authors to consider these suggestions in a minor revision. I encourage the authors to incorporate these minor suggestions, and I want to say congratulations for this interesting and original research.

We appreciate the comments and thank you for this complement.

General comments

  • Title is very long and complex. Please, simplify it. E.g remove after “ using…” or replace it for: “: A phenological approach”

We have made this change as suggested. The title is now: ”Characterizing growing season length of subtropical coniferous forests with a phenological model”.

  • Keywords could be improved. Some repetitions with the title (subtropical forest ecosystems, phenology model). I suggest add: modelling, ecophysiology, restoration…. The keywords must be presented in alphabetical order.

We have made this change as suggested. The key words now do not repeat any words in the title.

  • Introduction: An international context is lacking about climate change, and link with prescribed burning must be improved.

We have added two sentences to the Introduction that link phenology changes to global climate change feedbacks. This revision can be found on lines 42-59 of the “Track Changes” version of our revised manuscript. We have also added two additional sentences in the Introduction that discuss the importance of prescribed fire, and its interaction with climate change. This revision can be found on lines 102-117 of the “Track Changes” version of our revised manuscript.

  • Table 1-: 1 or 2 decimals no both

We have made this change as suggested, adding a decimal to the water holding capacity. Thus, DBH and water holding capacity are now one decimal place. We have kept the LAI, NDVI, and EVI with 2 decimal places, as these numbers tend to be one order of magnitude smaller than the other values in the table. While the number of decimal places across values are not consistent, we believe this is necessary to provide the resolution for these values. We also noted the units of water holding capacity, which were previously omitted.

  • Discussion: Make a new section (small) to talk about management implications.

We agree that this information is important to add to the paper. We have added a section to the Discussion (4.6) that discusses management implications of this work. This revision can be found on lines 526-538 of the “Track Changes” version of our revised manuscript.

Reviewer 2 Report

The submitted manuscript entitled Characterizing variation in growing season length of subtropical coniferous forests using a photosynthesis-based phenology model presents results of very interesting research which will be very helpful in for understanding the functioning of trees in changing climate conditions. The study was conducted on mature subtropical longleaf pine forests in USA. It should be noted here that this is an ecosystem sensitive to the increased risk of wild fires due to the forecasted periods of summer droughts, heat waves and their increased frequency.

Generally, the manuscript is very well written. I have only minor comments to main text listed in specific comments.The Introduction presents the research problem well. The aims of research and hypotheses are clear and logical. The Materials and methods are described in details. Also Results section is well written, figures and tables are legible and well prepared. Nice Discussion. Conclusion are reasonable and well supported by the obtained results.

In my opinion, afer minor technical corrections the submitted manuscript will be suitable for publication in Forests journal.

Specific comments:

L.34-35: The task of keywords is to provide supplementary information that is not included in the title and better positioning of the publication in web browsers. If your keywords are also in the title, they lose that function. It is worth considering changing some of them.

L.45: phenological activity -> it sounds strange.

L.564-565: The number of decimal places for p-values should be provided consistently for all results.

Nice paper! Congratulations!

Author Response

Our replies to Reviewer 2 are below each comment, in italics.

The submitted manuscript entitled Characterizing variation in growing season length of subtropical coniferous forests using a photosynthesis-based phenology model presents results of very interesting research which will be very helpful in for understanding the functioning of trees in changing climate conditions. The study was conducted on mature subtropical longleaf pine forests in USA. It should be noted here that this is an ecosystem sensitive to the increased risk of wild fires due to the forecasted periods of summer droughts, heat waves and their increased frequency.

Generally, the manuscript is very well written. I have only minor comments to main text listed in specific comments.The Introduction presents the research problem well. The aims of research and hypotheses are clear and logical. The Materials and methods are described in details. Also Results section is well written, figures and tables are legible and well prepared. Nice Discussion. Conclusion are reasonable and well supported by the obtained results.

In my opinion, afer minor technical corrections the submitted manuscript will be suitable for publication in Forests journal.

Specific comments:

L.34-35: The task of keywords is to provide supplementary information that is not included in the title and better positioning of the publication in web browsers. If your keywords are also in the title, they lose that function. It is worth considering changing some of them.

We have made this change as suggested. The key words now do not repeat any words in the title.

L.45: phenological activity -> it sounds strange.

We have changed this to “the timing of phenological events”. This revision can be found on line 58 of the “Track Changes” version of our revised manuscript.

L.564-565: The number of decimal places for p-values should be provided consistently for all results.

We have made this change as suggested. The number of decimals has been adjusted for both Tables A2 and A3. These revisions can be found on lines 619-630 of the “Track Changes” version of our revised manuscript.

Nice paper! Congratulations!

We appreciate the comments and thank you for this complement.